# Spaceborne potential for examining taiga-tundra ecotone form and vulnerability

**P. M. Montesano[1,2], G. Sun[2,3], R. O. Dubayah[3], and K. J. Ranson[2]**

[1] Science Systems and Applications, Inc., Lanham, MD, 20706, USA

[2] Biospheric Sciences Laboratory, NASA Goddard Space Flight Center, Greenbelt, MD 20771, USA

[3] University of Maryland, Department of Geographical Sciences, College Park, MD 20742, USA

Correspondence to: P. M. Montesano (paul.m.montesano@nasa.gov)

**Abstract**

In the taiga-tundra ecotone (TTE), site-dependent forest structure characteristics can influence the subtle and heterogeneous structural changes that occur across the broad circumpolar extent. Such changes may be related to ecotone form, described by the horizontal and vertical patterns of forest structure (e.g., tree cover, density and height) within TTE forest patches, driven by local site conditions, and linked to ecotone dynamics. The unique circumstance of subtle, variable and widespread vegetation change warrants the application of spaceborne data including high-resolution (< 5m) spaceborne imagery (HRSI) across broad scales for examining TTE form and predicting dynamics. This study analyzes forest structure at the patch-scale in the TTE to provide a means to examine both vertical and horizontal components of ecotone form. We demonstrate the potential of spaceborne data for integrating forest height and density to assess TTE form at the scale of forest patches across the circumpolar biome by (1) mapping forest patches in study sites along the TTE in northern Siberia with a multi-resolution suite of spaceborne data, and (2) examining the uncertainty of forest patch height from this suite of data across sites of primarily diffuse TTE forms. Results demonstrate the opportunities for improving patch-scale spaceborne estimates of forest height, the vertical component of TTE form, with HRSI. The distribution of relative maximum height uncertainty based on prediction intervals is centered at ~40%, constraining the use of height for discerning differences in forest patches. We discuss this

uncertainty in light of a conceptual model of general ecotone forms, and highlight how the
uncertainty of spaceborne estimates of height can contribute to the uncertainty in identifying TTE
forms. A focus on reducing the uncertainty of height estimates in forest patches may improve
depiction of TTE form, which may help explain variable forest responses in the TTE to climate
change and the vulnerability of portions of the TTE to forest structure change.

## 1    Introduction

### 1.1    TTE vegetation structure and processes

The circumpolar biome boundary between the boreal forest and arctic tundra, also known as
the tree-line, the forest-tundra ecotone, or the taiga-tundra ecotone (TTE), is an ecological
transition zone covering > 1.9 million km$^2$ across North America and Eurasia (Payette et al, 2001;
Ranson et al., 2011). This ecotone is among the fastest warming on the planet (Bader, 2014). The
location, extent, structure and pattern of vegetation in the TTE influences interactions between the
biosphere and the atmosphere through changes to the surface energy balance and distribution of
carbon (Bonan, 2008; Callaghan et al., 2002a). These TTE vegetation characteristics also affect
local and regional arctic and sub-arctic biodiversity (Hofgaard et al., 2012) and are controlled by
a variety of factors that are scale-dependent (Holtmeier and Broll, 2005). At local scales the spatial
configuration of trees is determined largely by site-level heterogeneity in hydrology, permafrost,
disturbance, topography (aspect, slope, elevation), land use and the geomorphologic conditions
associated with each (Dalen and Hofgaard, 2005; Danby and Hik, 2007; Frost et al., 2014; Haugo
et al., 2011; Holtmeier and Broll, 2010; Lloyd et al., 2003).
North of the Kheta River in central Siberia (e.g., 71.9°N 101.1°E), the TTE exhibits a change
in forest structure across a gradient of open canopy (discontinuous) forest from south to north. In
this region, latitude coarsely controls TTE forest structure characteristics, which feature a general
decrease in height and cover from south to north, as well as a variety of spatial patterns of trees
(Holtmeier and Broll, 2010). These structural characteristics influence a range of TTE
biogeophysical and biogeochemical processes in a number of ways. Forest structure provides
clues as to the extent of sites with high organic matter accumulation and below-ground carbon
pools (Thompson et al., 2016). Recent work notes that rapid growth changes individual tree forms,
thus altering recruitment dynamics (Dufour-Tremblay et al., 2012).  Height and canopy cover of
trees and shrubs affect site-level radiative cooling, whereby larger canopies increase nocturnal
warming and influence regeneration (D'Odorico et al., 2012).  Such tree height and canopy controls
over the transmission of solar energy have been well documented (Davis et al., 1997; Hardy et al.,
1998; Ni et al., 1997; Zhang, 2004). The height and configuration of vegetation also partly
influences permafrost by controlling snow supply, creating heterogenuous ground and permafrost
temperatures (Roy-Léveillée et al., 2014).  Accounting for vegetation heterogeneity in schemes
addressing surface radiation dynamics helps address the effects on rates of snowmelt in the boreal
forest (Ni-Meister and Gao, 2011).  Modeling results support the importance of tree heights on
boreal forest albedo, which is a function of canopy structure, the snow regime, and the angular
distribution of irradiance (Ni and Woodcock, 2000).  Better representation of vegetation height
and cover are needed to improve climate prediction and understand vegetation controls on the
snow-albedo feedback in the high northern latitudes (Bonfils et al., 2012; Loranty et al., 2013).
Furthermore, the structure of vegetation in the TTE helps regulate biodiversity, where the
arrangement of groups of trees provides critical habitat for arctic flora and fauna (Harper et al.,
2011; Hofgaard et al., 2012).
**1.2   A conceptual model of the TTE: forest patches, ecotone form and the link to structural**
**vulnerability**
The TTE, and other forest ecotones, can be conceptualized as self-organizing systems
because of the feedbacks between the spatial patterns of groups of trees and associated ecological
processes (Bekker, 2005; Malanson et al., 2006).  In this conceptual model groups of trees with
similar vertical and horizontal structural characteristics can be represented as forest patches.  These
patches have ecological meaning, because they reflect similar site history and environmental
factors.  At a coarser scale, these patterns and structural characteristics of TTE forest patches have
been conceptualized with a few general and globally recognized ecotone forms (Harsch and Bader,
2011; Holtmeier and Broll, 2010).  In the TTE, these general ecotone forms (diffuse, abrupt, island,
krummholz) reflect the spatial patterns of forest patches that are described by the horizontal and
vertical structural characteristics of trees (e.g. canopy cover, height and density), and have different
primary mechanisms controlling tree growth.
The variation in ecotone form may help explain differing rates of TTE forest change across
the circumpolar domain.  These forms tend to vary with site factors, which may partly control the
heterogeneity of change seen across the circumpolar TTE (Harsch and Bader, 2011; Lloyd et al.,
2002).  Further investigation is needed into the link between observed changes in vegetation, their
pattern, and local factors that may control these changes (Virtanen et al., 2010).  Epstein et al. 2004
provide a synthesis of how TTE patterns and dynamics are linked, and explain that a better
understanding of vegetation transitions can improve predictions of vegetation sensitivity.  Their
observations provide a basis for the inference that TTE structure is most susceptible to
temperature-induced changes in its structure where its structure is temperature-limited.  Thus, the
structural vulnerability of the TTE may be broadly defined as the susceptibility of its vegetation
structure to changes that result in shifts in its geographic position and changes to its spatial pattern
of trees.  Vulnerable portions of the TTE are areas most likely to experience changes in forest
structure that alter TTE structural patterns captured by forest patches and described by ecotone
form.
**1.3   Towards identifying TTE form: spaceborne data integration, scaling and the**
**uncertainty of TTE structure**
Spaceborne remote sensing data may facilitate identifying TTE form and linking it to local
site factors and structural vulnerability (Callaghan et al., 2010; 2002b; Harsch and Bader, 2011;
Kent et al., 1997).  They way in which spaceborne data is integrated and scaled may be a key part
of identifying structural patterns and TTE form.  Fine-scale data can resolve individual trees that,
when grouped to patches, may reveal ecotone forms (Danby and Hik, 2007; Hansen-Bristow and
Ives, 1985; Hofgaard et al., 2012; 2009; Holtmeier and Broll, 2010; Mathisen et al., 2013).
Without resolving groups of individual trees, coarse studies of the land surface may misrepresent
ecotone form, be less frequently corroborated with ground data, and disguise the structural
heterogeneity of discontinuous forests.  In a TTE landscape this structural heterogeneity is critical
for understanding biodiversity, biogeochemical and biophysical characteristics such as carbon
sources, sinks and fluxes, permafrost dynamics, surface roughness, albedo, and evapotranspiration
(Bonan, 2008).  Furthermore, understanding at a fine-scale where the TTE is likely to change may
improve understanding of the potential effects of changing TTE structure on these regional and
global processes.
A forest patch approach to the integration of multi-resolution remote sensing data may
mitigate data scaling issues with regard to forest structure estimates. One example of mitigation
is the misrepresentation of forest structure that arises with the sole use of coarse data. Medium-
resolution sensors such as Landsat and ALOS may not be suited for identifying the patch
boundaries at the resolution required to study TTE structure. However, their spectral or backscatter
information may still have value for predicting patch characteristics when combined with the
spatial detail of high resolution spaceborne imagery (HRSI) to define patch boundaries. Such an
approach integrates coarser data into an analysis while maintaining the spatial fidelity of feature
boundaries. Furthermore, a patch-level analysis helps attenuate high frequency noise in image
data. For example, ALOS PALSAR backscatter has significant pixel-level speckle (Le Toan et
al., 2011; Mette et al., 2004; Shamsoddini and Trinder, 2012) which, when grouped with
coincident HRSI patch boundaries, can be averaged to reduce the noise and quantified further with
a variance estimate.
In particular, data integration and scaling may also help mitigate the uncertainty of spaceborne
estimates of vertical structure in discontinuous TTE forests. A spaceborne assessment of forest
structure from individual active sensors across a gradient of boreal forest structure shows broad
ranges of uncertainty at plot-scales (Montesano et al., 2014a; 2015). These plot-scales studies
provide an indication of the scale at which TTE structure changes. A spaceborne remote sensing
approach that identifies forest patch boundaries with HRSI may provide insight into TTE structural
characteristics that are indicative of general ecotone forms at scales that are dictated by the
variation of TTE forest structure itself. As such, a patch-based approach to capturing forest height
and forest height uncertainty in the ecotone capitalizes on the added value that estimates of
horizontal structure may provide for reducing uncertainties in estimates of vertical structure from
remote sensing.
An evaluation of forest structure uncertainty serves the long-term goal of monitoring change
over time and between sites, as well as distinguishing the portions of the TTE that are vulnerable
to changes in forest height, cover or density from those whose structure is more resilient, and the
rates associated with these changes (Epstein et al., 2004). The spatial patterns of this structural
vulnerability will help models predict the consequences of TTE structural change on regional and
global processes.
This work examines the uncertainty of mapped forest patch heights using a spaceborne remote
sensing data integration approach. We map forest patches with HRSI data (<5 m) to spatially
assemble a medium spatial resolution (5 m - 50 m) suite of measurements from multi-spectral
optical and SAR with light detection and ranging (LiDAR) samples to estimate and model forest
height and its uncertainty by forest patch. We discuss the implication of this uncertainty for both
identifying TTE form and predicting dynamics, with regard to separating identifying portions of
the TTE whose forest structure is vulnerable to temperature-induced changes.

**2    Methods**
**2.1    Study area & ground reference data**
Our study area encompasses a region of the TTE in northern Siberia in which we identified
forest patch mapping sites and incorporated existing calibration and validation field plot and stand
data. The region is subject to a severe continental climate, generally exhibits a gradient in tree
cover from discontinuous to sparse, features elevations generally < 50 m.a.s.l., and is underlain
with continuous permafrost (Bondarev, 1997; Naurzbaev et al., 2004). The forest cover,
exclusively *Larix gmelinii* across all mapping, calibration and validation sites, exists at the climatic
limit of forest vegetation, coinciding closely with the July 10°C isotherm (Osawa and Kajimoto,
2009). Tall shrubs, including *Alnus sp.*, *Betula sp.*, and *Salix sp.*, and dwarf shrubs (e.g. *Vaccinium*
*sp.*), occur along with sedge-grass, moss and lichen ground covers.
The mapping sites are primarily situated on the Kheta-Khatanga Plain, north of the Kheta
River, which is a tributary of the Khatanga River flowing north into the Laptev Sea. One site,
which sits just south of the Novaya River on the Taymyr Peninsula, includes a portion of Ary-
Mas, the world's northernmost forest (Bondarev, 1997; Kharuk et al., 2007; Naurzbaev and
Vaganov, 2000). Mapping sites were chosen based on the presence of cloud-free multispectral
and stereo pair data from HRSI available in the Digital Globe archive, and presence of patches of
forest cover (Neigh et al., 2013). We visually interpreted HRSI to identify sites in this portion the
TTE where forest cover was discontinuous and where forest patches exhibited diffuse, abrupt or
island ecotone patch forms.

Ground reference sites were derived from two sources. The first consisted of individual tree

measurements at circular plots (15 m radius) coincident with spaceborne LiDAR footprints while
the second comprised stand-level data specific to *Larix gmelinii* across a broader central Siberian
region. The plot data, collected during an August 2008 expedition to the Kotuykan and Kotuy
Rivers, were used as either calibration or validation data in this study (Montesano et al., 2014b).
Measurements were collected of tree diameters at breast height (DBH, 1.3 m) and tree heights
(clinometers for 97% of trees and tape measurement for 3%) at plots coincident with spaceborne
LiDAR footprints. The data used for this study included DBH for all tree stems with DBH >3 cm
(±0.1 cm) and corresponding tree heights for each tree in each plot. These plot data, representing
a range of discontinuous *Larix gmelinii* forest conditions found across northern Siberia excluding
prostrate tree forms, were supplemented with the stand data reported in Bondarev (1997). Shrub
structure was not considered in this study.

The forest mapping and ground reference sites do not spatially coincide. This study

examines the TTE on the Kheta-Khatanga Plain which exhibits a range of TTE forms, where the
TTE covers a broader area, and where we had access to both stereo and multispectral HRSI data.
While not spatially coincident, our ground reference sites characterize very similar forest
conditions to those in the mapping sites. The main difference is that the ground reference sites
feature an ecotone that is compressed, covering a smaller area due to topography, relative to the
mapping sites. The type and structure of the *Larix gmelinii* forests is consistent across the broader
region (Bondarev, 1997). The geographic footprints of all mapping sites for which forest patches
were examined, as well as the general locations of Kotuykan/Kotuy ground reference sites, are
shown in Figure 1.
**2.2   Spaceborne data acquisition and processing**

A suite of spaceborne remote sensing datasets were used in this study to delineate forest

patch boundaries, assign forest patches with remote sensing image pixel values, and predict forest
patch height. Table 1 lists the individual data sets along with their period of acquisition. These
data were collected within ~8 year period (2004 - 2012) across sites during which, based on visual
inspection of HRSI, there were no signs of disturbance from fires, and for which the rate of tree
growth is likely well below that which would be detectable from spaceborne data in that time
interval. The data include spaceborne LiDAR data from the ICESat satellite's Geoscience Laser
Altimeter System (GLAS) and image data from passive optical Landsat-7 ETM and Worldview-1
& -2, and synthetic aperture radar (SAR) from ALOS PALSAR.

2.2.1    Spaceborne LiDAR data

The spaceborne LiDAR data from GLAS featured ground footprint samples ~60 m in
diameter (the actual footprint is an ellipse) of binned elevation returns of features within each
footprint.  These data provided ground surface elevation samples as described in a previous study
(Montesano et al., 2014b).   The set of GLAS data coincident with the DSM of the study sites was
filtered in an effort to remove LiDAR footprints for which within-footprint elevation changes
precluded capturing heights of trees generally less than 12 m tall.  The GLAS footprints used
satisfied the following conditions; (1) the set of coincident DSM pixels had a standard deviation
≤ 5 m, (2) the length of the LiDAR waveform was ≤20 m, and (3) the difference between the
maximum and minimum DSM values within a 10 m radius of the GLAS LiDAR centroid was ≤
25 m.  This radius helped remove footprints for which there was a broad range of DSM values
near the footprint centroid, indicative of terrain slope that would likely interfere with forest height
estimation.

2.2.2    Spaceborne Image data

Spaceborne image data covering the full extent of each study site that were resampled from
their original un-projected format during a re-projection into the Universal Transverse Mercator
coordinate system (zone 48).  The images were either medium (25 m-30 m pixels) or high (<5 m
pixels) resolution.   The medium resolution spaceborne imagery included a Landsat-7 ETM
multispectral cloud-free composite and vegetation continuous fields tree cover (VCF) products
and ALOS PALSAR tiled yearly mosaics (2007 - 2010) (Hansen et al., 2013; Shimada et al., 2014).
The four ALOS PALSAR yearly mosaics were processed into an average temporal mosaic of dual
polarization (HH and HV) backscatter power.  The high resolution data consisted of HRSI
multispectral (Worldview-2 satellite) and panchromatic (Worldview-1 satellite) data acquired
from the National Geospatial Intelligence Agency via the NextView License agreement between
Digital Globe and the US Government (Neigh et al., 2013).
This HRSI was processed in accordance with Montesano et al. (2014) to generate a digital
surface model (DSM) of elevations for each study site using the NASA Ames Stereo Pipeline
software (Moratto et al. 2010; Montesano et al., 2014b).  In addition to DSM generation, the HRSI
data were processed to compute three additional image layers that were used to delineate and
assign forest patches with the mean and variance of corresponding image pixel values.  The steps
below describe the processing of the 3 additional layers:
*NDVI image*: We computed a normalized difference vegetation index (NDVI) layer to create
a mask separating areas of vegetation from non-vegetation within each mapping site.  This widely
used algorithm was based on the near-infrared (NIR) and red channels of the multispectral HRSI
(*[NIR – red] / [NIR + red]*).  This NDVI calculation, based on uncalibrated digital number values
of image pixels, supported the objective of classifying forest structure patterns rather than
maintaining the fidelity of reflectance characteristics.
*Panchromatic image roughness*:  This roughness data was based on the textural
characteristics of each site's panchromatic HRSI.  Image roughness/texture information is useful
for examining horizontal forest structure, a component of which is tree density (e.g., Wood et al.,
2012; Wood et al., 2013).  We computed image roughness using the output layers from the bright
and dark edge detection (described in Steps 10-12 of Table 2 in Johansen et al.) (Johansen et al.,
2014).  This image roughness derivation is resolution independent in that feature roughness can be
captured as long as those features are resolved in the imagery.  Here, we use ~60cm data to quantify
a signal from groups of *Larix gmelinii* trees.  The output from this roughness computation was a
single image layer showing increased brightness values corresponding to increasingly textured
surface features that is a result of the arrangement of trees across the landscape.
*Canopy roughness model*:  The second of two image roughness layers, a canopy roughness
model (CRM), was calculated from each DSM.  A low pass (averaging) filter (kernel size = 25 x
25) was applied to a version of the DSM that was resampled to decrease the spatial resolution by
a factor of 8.  The filtering generated a smoothed terrain elevation ($elev_{terrain}$) layer that removed
the elevation spikes from the discontinuous tree cover that is evident in the DSM.  This $elev_{terrain}$
layer was then resampled to the original spatial resolution.  Surface feature roughness was
computed as the difference between the DSM and $elev_{terrain}$, and were represented as heights above
$elev_{terrain}$.
**2.3 Forest masking, patch delineation and value assignment**
We analyzed forest structure at the study sites by masking forest area, delineating forest
patch boundaries and assigning these patches with remotely sensed data values in order to model
forest patch height. This delineation and value assignment framework used the segmentation
algorithms in Definiens Developer 8.7 (Benz et al., 2004). This framework modifies the multi-
step, iterative segmentation and classification procedure discussed in previous work (Montesano
et al., 2013). The central difference is that this approach uses exclusively data from HRSI to
identify a vegetation mask and refine it to create a forest mask. We applied a segmentation to this
forest mask to separate distinct forest patches, and then assigned those patches the mean and
standard deviation of pixel values from all coincident data.
Creating the forest mask was an iterative process that included segmentation and
thresholding of the NDVI and 2 roughness layers. The thresholds used to classify forest were
based on preliminary interpretation of the *Larix gmelinii* forest and non-forest areas in imagery
across all forest patch mapping sites. The goal of this preliminary exploratory work was to
understand the range of roughness and NDVI values associated with forest identified with visual
interpretation of the particular set of imagery used. This exploratory work identified thresholds
that were image independent and could be used in an automated patch classification protocol across
all sites. However, these thresholds are sensitive to the seasonality of vegetation and, likely, the
sun-sensor-target geometry at which the imagery was acquired. A detailed examination of the
trade-offs associated with threshold choices and forest mask results was not part of this work.
The preliminary vegetation mask, generated from the initial separation of vegetation and
non-vegetation within mapping sites, was based on an unsupervised contrast-based segmentation
of the NDVI layer. This first masking step was further modified with NDVI and image roughness
thresholding steps to compile a final forest mask. Next, we used both the panchromatic-derived
roughness layer and the DSM-derived CRM to capture vegetation roughness and modify the
preliminary vegetation mask. Thresholds were applied to these two roughness layers to create a
forest mask sub-category. First, forest was separated from non-forest based on a panchromatic
HRSI roughness threshold value = 5.5, where higher values represented rougher vegetation and
were classified as forest. Second, the forest mask was refined with information from the CRM. A
CRM threshold value = 1 was used to reclassify existing non-forest regions into the forest class.
In the final step of this iterative forest masking process, remaining non-forest areas with a mean
roughness > 3 and mean NDVI < 0.25 were classified as forest. This helped classify remaining
vegetation whose roughness value suggested forest vegetation, but whose NDVI value had initially
excluded them from this class.

The forest mask provided the extent for which a 2-step procedure separated distinct forest
patches before assigning patches with image values. First, this forest mask was divided to separate
portions of forest whose roughness values were > 2 standard deviations above the median
roughness value. Next, patches were broken apart according to surface elevation values provided
from each site's DSM. Patches were assigned with the mean and standard deviation of image pixel
values within the boundary of each patch. Patch area was calculated to exclude patches below the
minimum mapping unit of 0.5 hectares. The remaining patches coincident with LiDAR footprint
samples were assigned forest patch height values via the direct height estimation approach
discussed below.
**2.4    Predicting forest patch height directly at LiDAR footprints**

GLAS LiDAR sampling of forest canopy height provided a means to estimate average patch
canopy height through direct spaceborne height measurements. Where forest patches coincided
with LiDAR footprints from GLAS, the canopy surface elevation from the DSMs and the ground
elevation from either the DSMs or GLAS within a GLAS LiDAR footprint provided a sampling
of forest height within the patch. First, we applied the methodology presented in Montesano et al.
(2014b) to compile spaceborne-derived canopy height within GLAS LiDAR footprints and convert
those heights to plot-scale maximum canopy height with a linear model (Montesano et al., 2014b).
Finally, these plot-scale canopy height predictions from all GLAS LiDAR footprints within a given
patch were used to directly determine the mean predicted forest patch height and the mean height
error from the prediction interval of the canopy height linear model.
**2.5    Modeling forest patch height indirectly**

Canopy height predictions were made indirectly for forest patches without direct spaceborne
sampling of forest canopy height. This indirect method, used for the vast majority (~90%) of forest
patches > 0.5 ha across the study sites, involved (1) building a model from the set of forest patches
with GLAS LiDAR samples relating the predicted forest patch canopy height (response variable)
to patch values from the spaceborne image data summarized in Table 1 (predictor variables) and
(2) applying that model to predict forest patch canopy height for those patches with no direct
spaceborne height samples. These methods, described in Montesano et al. (2013) and Kellndorfer
et al. (2010), use the Random Forest regression tree approach for prediction (Breiman, 2001;
Kellndorfer et al., 2010; Montesano et al., 2013). This approach includes specifying both the
number of decision trees that are averaged to produce the Random Forest prediction and the
number of randomly selected predictor variables used to determine each split in each regression
tree. The result is a prediction model that is valid for the range of predictions on which the model
was built and reduces overfitting, or, the degree to which the prediction model is applicable to only
the specific set of input data.

**3    Results**
**3.1    Forest patch delineation and direct sample density**

The forest patch was the fundamental unit of analysis in this study for which forest height
was assigned either directly from spaceborne data at GLAS LiDAR footprints, or indirectly from
spaceborne data by means of empirical modeling with Random Forest. A representative example
of a group of forest patches characteristic of a diffuse forest structure gradient delineated within
the study area in shown in Figure 2. Across the 9 study sites, 3931 forest patches > 0.5 ha were
delineated based on NDVI, image roughness and DSMs all from the HRSI data. Of this total, 364
patches (9%) coincided with at least one GLAS LiDAR footprint at which a height sample was
computed and used in the direct estimation of patch canopy height (Figure 3a). The bimodal
distribution that features a peak in the number of forest patches ~1 ha in size is evidence of the
heterogeneous nature of forest cover in this region. The plots in Figure 3b group forest patches,
for which direct height estimates were made, into categories based on patch area. They show the
general distribution of sampling density of direct height estimates within these patches. All
patches with direct height samples featured a sampling density of < 3 samples ha$^{-1}$. The majority
(94%) of sampled patches had sampling densities < 0.5 samples ha$^{-1}$, of which most had patch
areas > 10 ha. Larger patches have lower sampling densities in part because of the irregular
arrangement of GLAS LiDAR tracks across the landscape.

## 3.2 Forest height calibration and validation


Forest height calibration and validation data were used to build and assess the empirical

model for direct spaceborne estimates of height. Figure 4a shows sites for which ground reference
calibration and validation data were collected. In Figure 4b, the corresponding distributions of
mean plot or stand height are shown for these sites. Measurements were collected in plots along
the Kotuykan River for this study (n = 69) and those from regionally coincident stands (n = 40) at
6 sites across northern Siberia from Bondarev (1997).

A portion of the Kotuykan/Kotuy River plots were used to calibrate (n = 33) the model used

to estimate spaceborne canopy height at plot-scales after Montesano et al. (2014b), which was
applied in the direct spaceborne estimation of forest patch height (Montesano et al., 2014b). The
remaining portion of the Kotuykan/Kotuy River plots (n = 36) and stands from Bondarev (1997)
(n = 40) served as independent validation of the distribution of forest patch heights derived from
direct spaceborne height estimation (Bondarev, 1997). Mean heights of forest patches, plots, and
stands were used to compare distributions of calibration and validation data because this was the
height metric that was consistently available across the set of forest patches, the calibration plots
and the validation plots and stands. The distributions in Figure 4c show the proportion of forest
patch heights for which direct spaceborne estimates of height were made. This distribution of
direct spaceborne estimates of forest patch heights is shown alongside the distributions of
individual tree measurements averaged across plots or stands from (1) the calibration plots in
Montesano et al. (2014b), (2) the remaining Kotuykan/Kotuy River validation plots, and (3) the
validation stands from Bondarev (1997).

## 3.3 Indirect forest patch height estimates


Indirect spaceborne estimates of forest patch heights were made for the majority of patches

examined. Maximum and mean forest heights were predicted for 91% of forest patches across the
study sites. Random Forest regression tree models for 5 sets of spaceborne data predictor variables
were used to estimate maximum and mean patch height indirectly for patches with no coincident
direct spaceborne height estimates. Figure 5 shows the residual standard error (RSE) and $R^2$ of
the best performing model (based on $R^2$) for each spaceborne data predictor set (a particular
combination of spaceborne data). The predictor set 'All' that included all spaceborne image data
layers identified in Table 1 explained > 60% of overall variation in modeled patch height. This
'All' model shows only incremental improvement over the model using only HRSI-derived
predictors. The Landsat & ALOS spaceborne variables explain < 40% of variation within the
modeled relationship between spaceborne predictors and patch height.
**3.4    Uncertainty of forest patch height estimates**
We assessed the best performing Random Forest model for indirectly estimating maximum
and mean forest patch heights. The best performing models were those from the 'All' predictor
sets, described above, where the number of predictor variables was 14 and 15, for maximum and
mean forest patch height, respectively. Assessments were based on model $R^2$ and RMSE for the
maximum and mean patch height models, where 50% of patches with direct height estimates from
which the indirect models were built were used for model training and 50% were used for model
testing. The results of a bootstrapping procedure to examine the distribution of $R^2$ and RMSE
from the Random Forest models applied to the set of testing data is shown in Figure 6a,b. The
plots show the bootstrapped distributions of best performing model $R^2$ and RMSE, and are overlain
with boxplots. The Random Forest models for maximum and mean patch height explain 61% (+/-
14% at 2 σ) and 59% (+/- 14% at 2 σ) of the variation with errors of 1.6 m (+/- 0.2 m at 2 σ) and
1.3 (+/- 0.2 m at 2 σ), respectively, where 2 σ represents the 95% confidence interval.
We computed 95% prediction intervals for patches receiving both direct and indirect height
estimates. These prediction intervals show the uncertainty associated with patch-level estimates
of both maximum and mean patch heights. Figure 7a shows these height estimates and prediction
intervals for all patches in this study across the continuum of patch sizes. Figure 7b shows the
relative prediction error, which was computed as the difference between the upper and lower
prediction interval range divided by the predicted height value.

**4    Discussion**
Recent work suggests that TTE form may reflect which portions of the TTE have forest
structure that is controlled primarily by temperature. With spaceborne remote sensing, various
TTE forms across broad extents can be identified by characterizing the horizontal and vertical
structure of trees. By identifying these forms, the controls of TTE forest structure may be inferred.
The ability to characterize horizontal and vertical structure is a precursor to both (1) distinguishing
one TTE form from another, and (2) identifying areas where TTE form suggests tree growth is
temperature limited. The intersection of such temperature limited TTE forms with regional
warming trends may point to areas where TTE forests are vulnerable to changes in its structure.
Our work demonstrates the potential from spaceborne remote sensing for depicting a key structural
characteristic of TTE form (height), and suggests where improvements are needed in order to
identify portions of the TTE vulnerable to warming-induced structural changes.
This study's site-scale approach to examining forest structure is an example of a way to
quantify the potential for change in forest structure and its effects on broader TTE dynamics. Such
detailed monitoring is needed to resolve both the variability in TTE forest structure at fine spatial
scales and the variability in structural responses to changes in environmental drivers that are
observed across the TTE. The high resolution delineation of forest patches at our study sites in
the TTE of northern Siberia demonstrates the detailed monitoring that is possible for examining
spatial patterns of forest structure across the circumpolar domain, because of the use of spaceborne
data. The forest patch height prediction intervals are estimates of the measurement error at the
forest patch scale that explain existing constraints for discerning TTE form linked to changes in
TTE forest structure.
We discuss the utility of the patch-based analysis, review the patch-level estimates of
uncertainty and then examine them in the context of a conceptual biogeographic model of TTE
forest structure presented in recent literature. Such a model helps clarify and focus spaceborne
approaches to examining characteristics of TTE forest structure and its vulnerability to structural
change.
**4.1 Patch-based TTE forest structure analysis**
The patch-based approach of remotely measuring TTE forest structure addresses the
imperative for site-scale detail of TTE vegetation, whereby individual trees can be resolved, while
acknowledging the influence of clusters of trees (patches) and their density on TTE attributes and
dynamics. This approach coarsens the data, reducing spatial detail. However, from a
biogeographic perspective, this reduction in detail is not arbitrary as are image pixel reductions
when images are coarsened by means of down-sampling. Rather, image features and ancillary
datasets inform the coarsening procedure, creating patch boundaries that are based on spectral and
textural characteristics of images as well as other landscape information. Polygonal patches,
particularly when vegetation patterns and heterogeneity are key landscape features, may be more
informative than pixels particularly for studies at fine scales. Furthermore, patches provide a
means to integrate remote sensing data across an area and extend sample measurements
(Kellndorfer et al., 2010; Lefsky, 2010; Montesano et al., 2013; van Aardt et al., 2006; Wulder and
Seemann, 2003; Wulder et al., 2007). We note that shrub structure was not accounted for in our
field data, and not directly addressed with our patch height analysis. However, it is likely that
signals from shrubs persisted in the forest mask used to estimate patch structure, and thus may be
incorporated into estimates of patch height and uncertainty.
**4.2    Forest patch height uncertainty**
There are four central results regarding the uncertainty of forest patch height across the study
area. The first two involve the sampling of canopy height within forest patches, while the last two
focus on its modeling. These local-scale results for the TTE are then contrasted with existing
global-scale estimates of forest height.
The way in which forest patch heights are sampled affects estimates. First, direct forest
patch height estimates from a combination of coincident GLAS LiDAR ground surface and HRSI
DSM-derived canopy elevations was made for ~9% of forest patches in the study area. Second,
the sampling density of these direct height estimates, driven by the sampling scheme of the
spaceborne LiDAR, is < 0.5 samples ha$^{-1}$ for 94% of sampled patches. This sampling density is
well below the critical density of 16 sample ha$^{-1}$ recommended for sampling forest biomass at the
1 ha plot-scale (Huang et al., 2013). These results suggest that the cost of increasing forest patch
sizes is a decrease in the density of direct height measurements. This is likely an artifact of the
GLAS sampling scheme, whose sampling is regular in the along-track direction (1 sample every
~170 m), but whose coverage of ground tracks was highly irregular across forested areas. Such a
sampling scheme likely increases patch height uncertainty, thus limiting the ability to discern
ecotone form.
The modeling of forest patch height provided some insight into what drives the prediction
of height and the associated uncertainty of predictions. First, the model that explained the most
variation included all remote sensing image data layers. However, this "all data" model showed
little improvement on that built from HRSI predictors. Furthermore, in the former, the most
important variables were from HRSI. These variables, NDVI and the standard deviation of the
canopy surface roughness, are indications of vegetation and its density within forest patches. This
suggests that the medium-resolution data from ALOS and Landsat products are not strong
predictors of vertical structure characteristics across the range of forest patch sizes identified in
the study area, and that without HRSI inputs, the heterogeneity of TTE forest structure at the scale
of its change across the ecological transition zone from forest to tundra is lost.

Second, the errors reported for the "all inputs" models predicting maximum and mean forest
patch height show forest patch height errors, including error uncertainty at $< 2$ m $\sigma$ (95%
confidence interval). However, the prediction intervals for these vertical structure metrics show
the uncertainty in the predictions at the patch-level of $\sim 40\%$. These patch-level prediction
intervals translate to a maximum patch height error of +/- 4 m for patches with maximum heights
of 10 m. These errors indicate that patches with maximum heights of 5 m and 10 m would be
statistically indistinguishable on the basis of height. This is a problem for identifying diffuse TTE
forms, for which forest patch and tree height is a key attribute, because these forms generally
features a gradual decrease in tree height and cover across portions of the ecotone where present.
Diffuse forms are the most likely type of general form to demonstrate treeline advance, where 80%
of diffuse ecotone sites examined in a meta-analysis show such treeline advance (Harsch et al.,
2009).

These local-scale uncertainties improve upon recent global-scale spaceborne maps of
vegetation height. These maps feature height uncertainties (RMSE) of $\sim 6$ m, which are expected
given that coarse-scale ($>500$ m) global maps of forest height aggregate many of these height
measurement samples across broad spatial extents (Lefsky, 2010; Simard et al., 2011). This
uncertainty can be the difference between the presence or absence of a forest patch in the TTE and
is therefore not suited for evaluating the link between TTE forest structure and heterogeneous
local-scale site factors. The height uncertainty of forest patches, $\sim 90\%$ of which have prediction
intervals less than $< 50\%$ of the predicted heights, improves the uncertainty and spatial resolution
of TTE forest height measurements. However, this study's primary benefit is in the fidelity of the
spatial extent of TTE forest patches. The scale of these patches are more appropriate than coarse,
global-scale estimates of forest structure for reporting site-specific forest structure estimates that
are critical for understanding forest characteristics at this biome boundary in flux.
**4.3    Improving the estimates of forest patch height**

Estimates of forest patch height need to be improved to distinguish important patch
characteristics. A potentially large source of uncertainty of patch height estimates may be
attributed to the limitation of the approach of using direct height estimates for calibration of the
indirect patch height prediction method. This approach for direct sampling of patch height, from
differencing canopy and ground surface elevations within LiDAR footprints, involves sampling a
very small portion of the overall patch. The assumption associated with delineating forest patches
is that each patch itself is a homogenous unit with similar tree structure characteristics throughout.
However, the extent to which this assumption holds was not examined. For patches with a high
degree of tree structure heterogeneity, a single direct sample of height may not be sufficient to
represent either maximum or mean patch heights. These data, when used to train a Random Forest
model, will degrade the modeled relationship of mean patch level image characteristics to patch
height, because the sample used to determine patch height might not be representative of actual
patch height.

There are two ways to address this source of uncertainty. The first is to accumulate more
direct samples of forest heights within a patch. This can be accomplished by collecting more
ground surface elevation estimates within forest patches. One way of doing this is with more
LiDAR samples. The LiDAR data collected after the launch of ICESat-2 should add to the existing
set of GLAS samples, contributing significantly to increasing ground surface elevation estimates
is forested areas, and adding enormous value to approaches that involve data integration from a
variety of sensors. More ground surface elevation estimates can also be made by improving the
way in which they are derived from HRSI DSMs. These improvements are needed because of
higher errors associated with HRSI DSM ground surface elevation estimates within forested areas
(Montesano et al., 2014b). Second, the homogeneity of forest patches can be improved by refining
algorithms associated with delineating forest patches. This could include decreasing patch size,
improving the canopy surface roughness algorithm (e.g., with tree-shadow fraction estimates), and
including multi-temporal HRSI to help separate surface features whose reflectance characteristics
differ throughout the growing season. These refinements may improve the modeling of forest
patch height and ultimately the ability to discern diffuse TTE forms.
**4.4    Spaceborne depiction of TTE form**
The conceptual model of ecotone forms presented by Harsch and Bader (2011) describes
form as a result of the relative dominance of different controlling mechanisms (Harsch and Bader,
2011). Only some of these mechanisms are primarily driven by climate. For the diffuse TTE
form, the primary controlling mechanism of this conceptual pattern is the growth-limitation of
trees, whereby tree-growth is driven by warming of summer or winter temperatures. This study
featured two key approaches for depicting diffuse TTE forms that may improve insight into the
vulnerability to climate warming of current TTE structure.
One key approach of this study involved integrating spatially detailed spaceborne
observations. This integration provided a means to simultaneously account for the horizontal and
vertical components of the spatial patterns of forest structure in the TTE that may help improve
depictions of the diffuse TTE form. Recent literature on the patterns of trees in the TTE explain
how tree density and height create varying forest patterns across the ecotone, that these patterns
are important because they may provide clues as to the dynamics of TTE forest structure, and that
they should be explored with detailed remote sensing (Bader et al., 2007; Harsch and Bader, 2011;
Holtmeier and Broll, 2007).
A second key approach aggregates the spaceborne estimates of horizontal and vertical
structure at the scale of forest patches. These patches provide a means to analyze the spatial pattern
of forest structure. This scaling is critical, because it facilitates a standardized approach to TTE
structure mapping that is appropriate for the broad spatial domain of the TTE while adhering to
requirements of site-specific forest structure detail. This helps to explore the biogeography of TTE
forest structure in the context of a conceptual model that highlights the importance of both TTE
tree density and height.
In this study, tree density is accounted for in an indirect manner with the delineation of
forest patches that use the horizontal structure captured with HRSI. This horizontal structure
manifests itself as image texture or the frequency of vegetation across a spatial extent, and may be
related to surface roughness, canopy cover or stem density, but a close examination of this
relationship was not part of this study. The patch-based approach for aggregating height
information was a means to break apart the forested portions of each site by reducing the
heterogeneity in horizontal structure. Essentially, the use of the roughness information derived
from HRSI helped establish a basis for the analysis of height by using it as a proxy for vegetation
density, and by expressing it as a contiguous patch that served as the fundamental unit by which
height was aggregated. This data integration should provide more information for discerning
diffuse TTE forms than individual assessments of either tree height or tree density.

The site-scale, patch-based treatment of the landscape is driven by two central needs. The
first is the need for site-level understanding of TTE vegetation structure characteristics. The second
is the need to understand the spatial patterns of trees across the landscape, because of the link
between vegetation patterns and ecological processes. This analytical approach should be
developed to more deeply explore the TTE vegetation patterns that variations in height and density
reveal, such as patch size, shape, landscape position, connectivity and spatial autocorrelation of
varying types of forest patches across the TTE as well as the association of such patterns with
permafrost and carbon flux dynamics.
**4.5   Implications for understanding TTE structure vulnerability**

Understanding the vulnerability of TTE structure is a key objective of research into expected
changes in the high northern latitudes (Callaghan et al., 2002a). Multiple lines of evidence indicate
that vegetation changes are occurring in the TTE, and that these changes are heterogeneous across
the circumpolar domain. The most rapid TTE vegetation responses to climate change will occur
where climate is the main factor controlling TTE vegetation (Epstein et al., 2004). This suggests
that TTE structure is most vulnerable at sites both controlled by, and undergoing changes in,
climate. Currently, the reported patch-level forest height uncertainty constrains the identification
of the portions of the TTE that are most vulnerable to forest structure change. However, this
spaceborne approach framed by the conceptual model of TTE form provides a clear directive for
near-term work of examining the biogeography of forest structure in the TTE, and understanding
and forecasting vegetation responses in the TTE based on the susceptibility to structural changes
(i.e. vulnerability) that these general patterns of forest structure suggest.
It is unlikely to derive the dominant mechanisms controlling TTE forest structure directly
from remote sensing. However, these mechanisms may be inferred from remotely sensed TTE
form. Depictions of diffuse TTE forms, resolved with improved maps of TTE patterns that
incorporate forest patch height estimates, may provide evidence as to the general mechanisms that
give rise to these diffuse forms (e.g. temperature-limited growth). Mapped TTE patterns, i.e. TTE
form, would be useful for examining ecosystem dynamics in the high northern latitudes. These
maps could be integrated with topographic, hydrologic, permafrost and other climate data to
suggest a gradient of TTE structure vulnerability. They would (1) provide information on the
patterns of environmental variables that are the dominant drivers of tree growth, (2) provide insight
into the influence of TTE structural changes on biodiversity (Hofgaard et al., 2012), and (3) inform
plant community and forest gap models that combine temperature, soil and disturbance data to
examine the drivers of vegetation structure and forecast its potential for change in the TTE (Epstein
et al., 2000; Xiaodong and Shugart, 2005). For example, understanding TTE form in areas where
vegetation structural changes have been noted may help explain the variability of structure change.
Furthermore, these depictions could also contribute to spatially explicit site index information in
ecosystem process models to help account for the variability in predictions of TTE forest structure
dynamics across the circumpolar domain. This will aid long-term forecasting by suggesting the
most likely sites, at fine scales, for changes to vegetation-disturbance feedbacks and the extent to
which biogeophysical interactions may shift (e.g., vegetation effects on surface albedo). The
vulnerability of TTE structure to temperature-induced change is one of many factors that may alter
ecological processes in the high northern latitudes.

**5    Conclusions**
The vertical component of TTE form, maximum and mean forest patch height, as derived
from a specific suite of spaceborne sensors at sites in northern Siberia, has an uncertainty of ~40%.
With this uncertainty, forest patches with maximum heights of 5 m and 10 m are statistically
indistinguishable on the basis of height. Height is a key attribute of the diffuse TTE forms, which
generally feature a gradual decrease of height and tree density across the ecotone and are the most
likely form to demonstrate treeline advance. Differences in the heights of forest patches are a
central feature of the diffuse TTE form where significant structural changes have been observed.
These differences suggests that improving the remote sensing of patch height will provide a key
variable needed for examining TTE forest structure. The conceptual model of TTE form should
continue to guide the application of a patch-based, multi-sensor spaceborne data approach because
of its potential for aggregating and scaling information provided by the structural patterns of
groups of forest patches across the full TTE domain. Such patterns may help infer which portions
of the TTE are most vulnerable to temperature-induced structural changes.

## 6 Acknowledgements

The use of trade names is intended for clarity only and does not constitute an endorsement of any
product or company by the federal government.

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

**8    Figures**

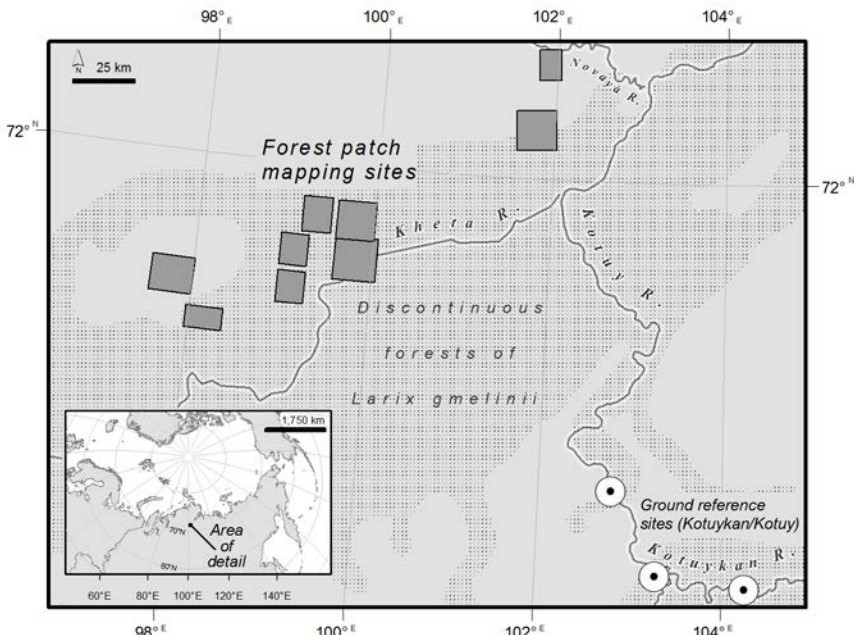


Figure 1. The study area in northern Siberia showing the 9 forest patch mapping sites (boxes) and
the ground reference sites along the Kotuykan River (circles) at which individual tree height
measurements in circular plots coincident with spaceborne LiDAR footprints were collected.


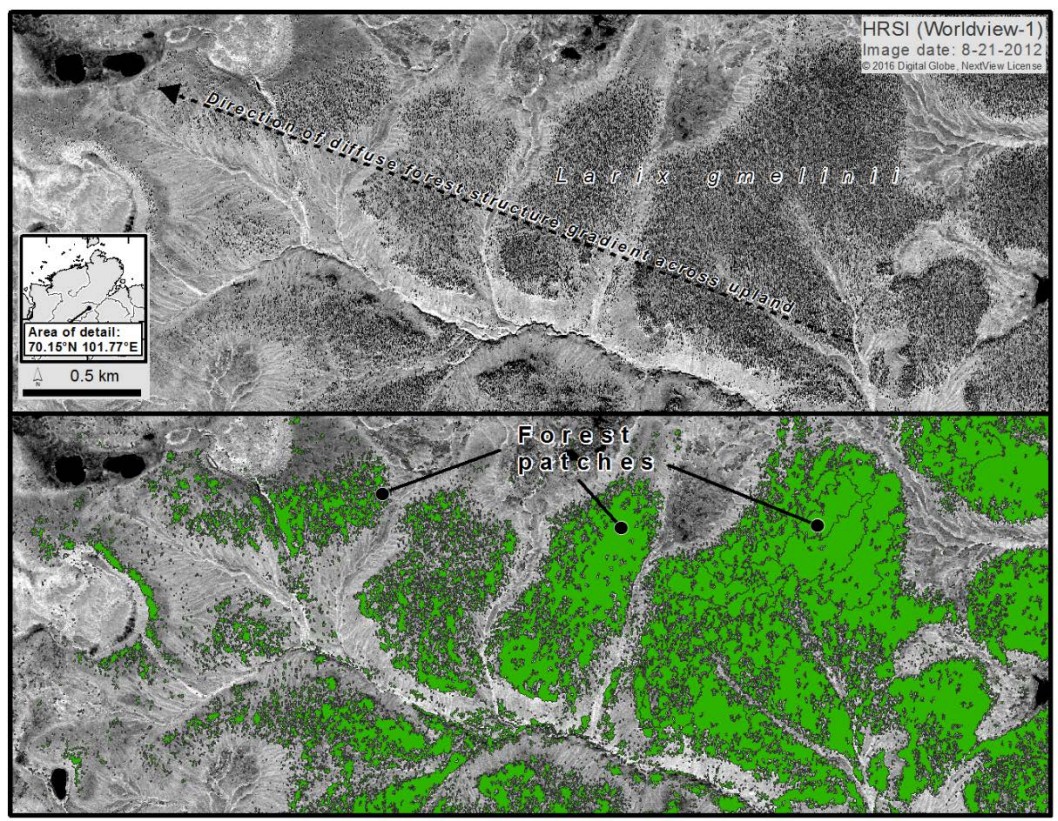

Figure 2. A representative example of forest patches showing a diffuse forest structure gradient of *Larix gmelinii* across an upland site delineated from HRSI. The top image shows a subset of a Worldview-1 panchromatic image from 8/21/2012 in one of the forest patch mapping sites. The bottom image shows the same subset with forest patches overlaid (green with gray outline).

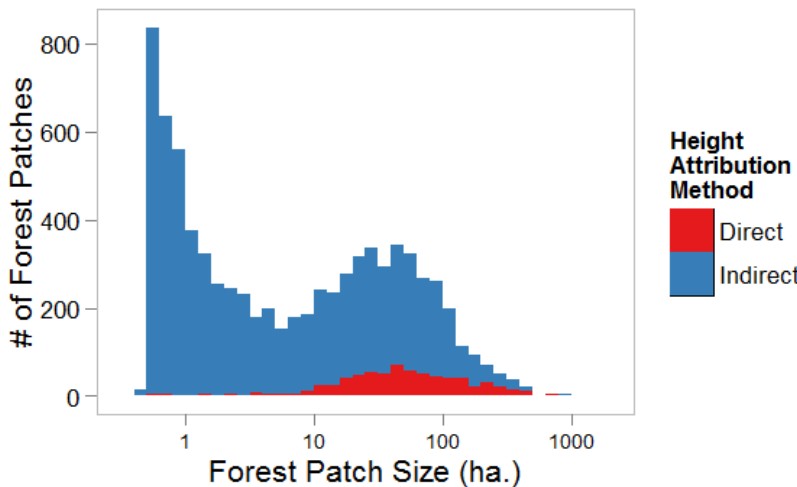


(a)


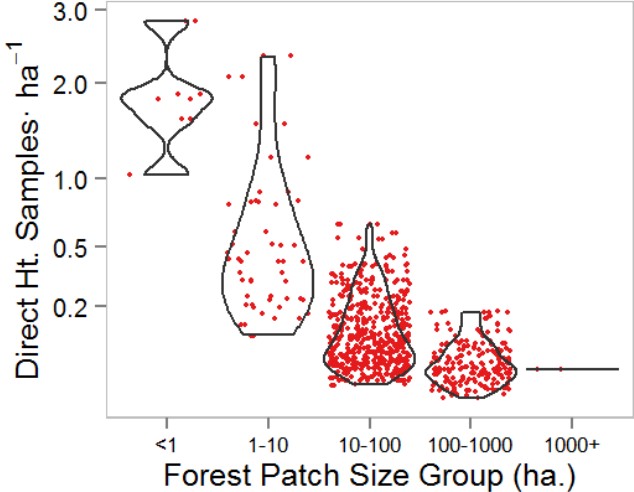

(b)
Figure 3. (a) The distributions of forest patch size in hectares according to height attribution
method. (b) The distribution of direct height sample density (shown as violin plots) for each forest
patch size group, overlain with dots representing individual patches (red).

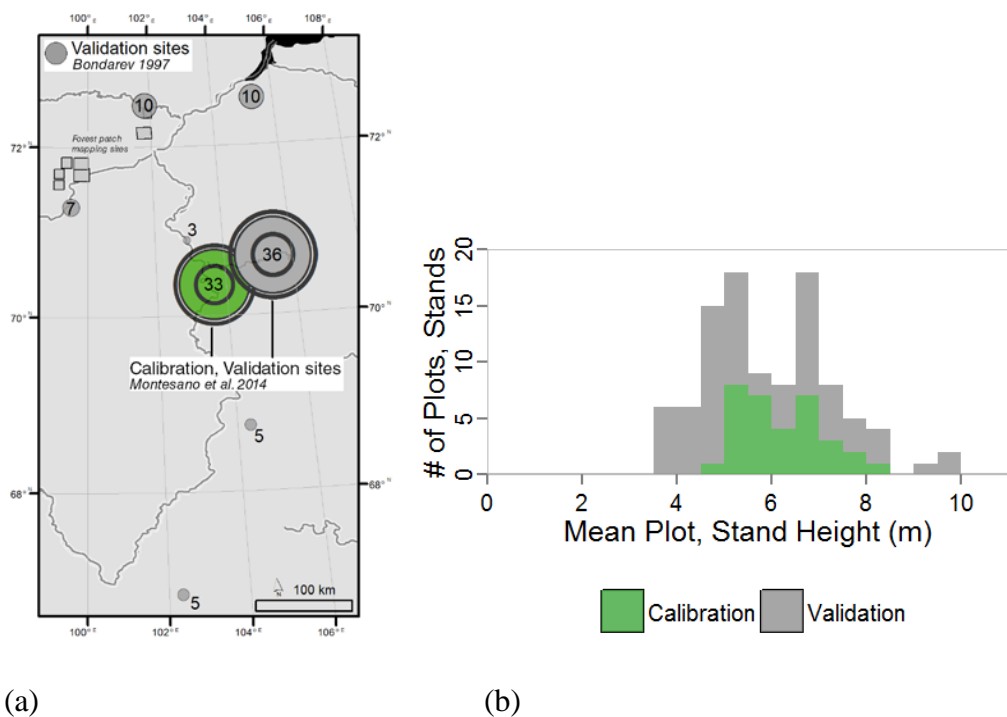


(a)                                                    (b)

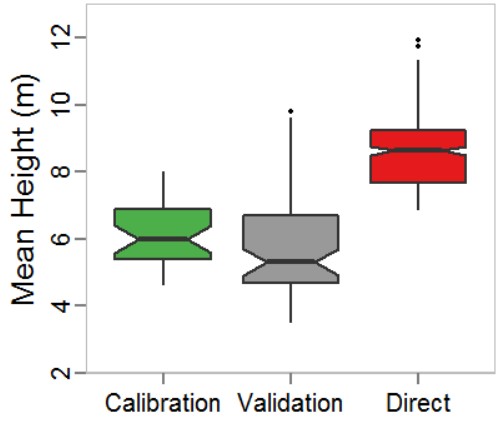


(c)
Figure 4. (a) Map of locations of calibration (green) and validation (grey) sites in northern Siberia
with the number of stands or plots associated with each site. The circles representing general site
locations are sized according to the number of stands. (b) Histogram of mean plot and stand heights
from calibration and validation data. (c) Comparison of the distribution of mean height of
calibration and validation plots and stands with that of forest patches heights from direct estimates.
Notched boxplots showing the 25th, 50th, and 75th percentiles of mean height as horizontal lines
and 1.5 times the inter-quartile range as vertical lines. Notches roughly indicate the 95%
confidence interval for the median.

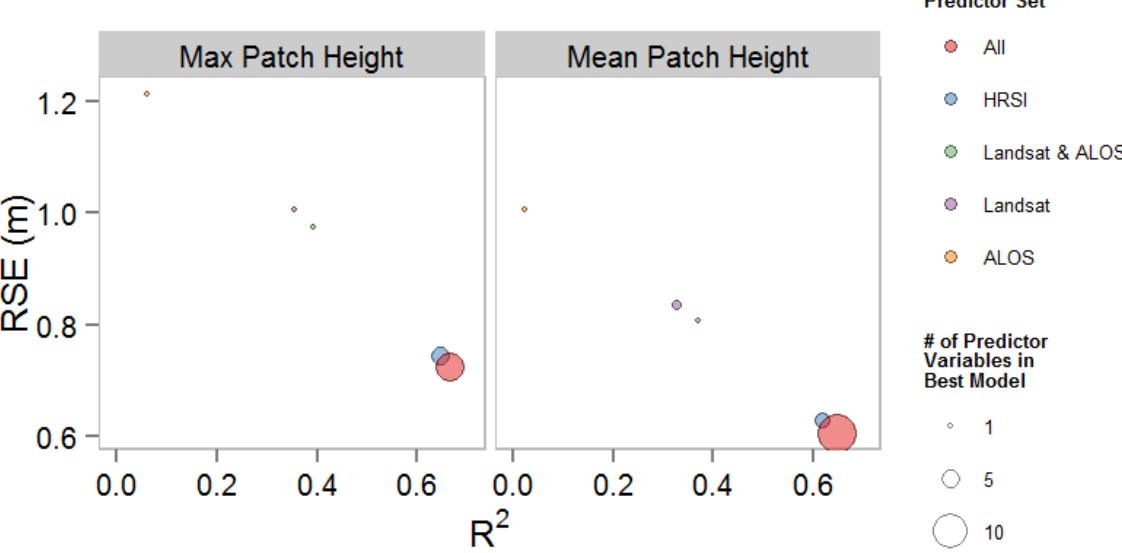


Figure 5. Results from Random Forest indirect forest patch height estimation for 5 spaceborne data
predictor sets.


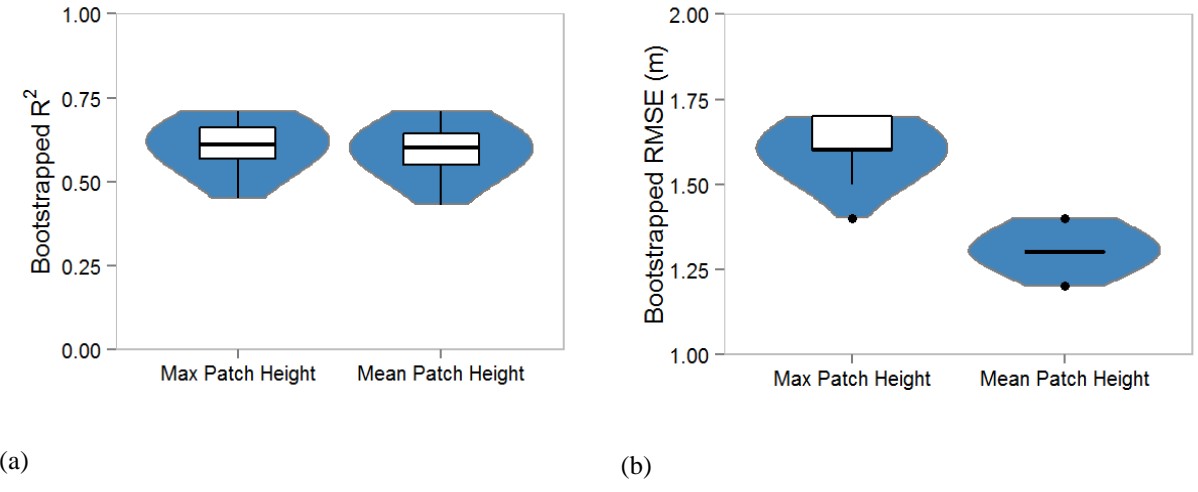

(a)                      (b)

Figure 6. The bootstrap-derived distributions (shown as violin plots, blue) of the Random Forest
model's (a) $R^2$ and (b) RMSE for the indirect forest patch height prediction method whereby all
spaceborne variables were used to predict maximum and mean forest patch height. Boxplots
(white) show the 25th and 75th percentiles (lower and upper lines), median (dark line), and 1.5 *
the inter-quartile range (whiskers). Data beyond the whiskers are shown as points.

(a)

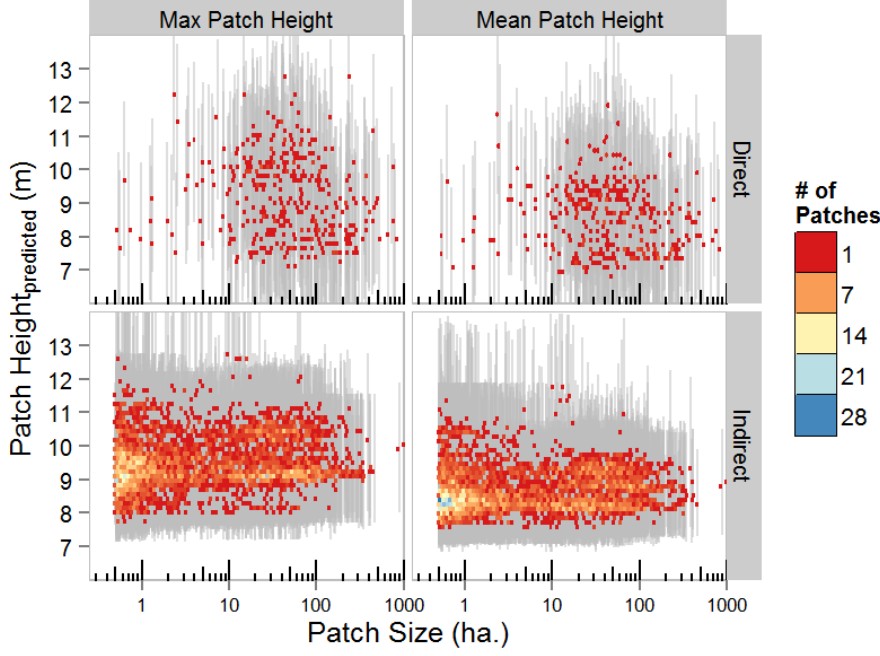

(b)

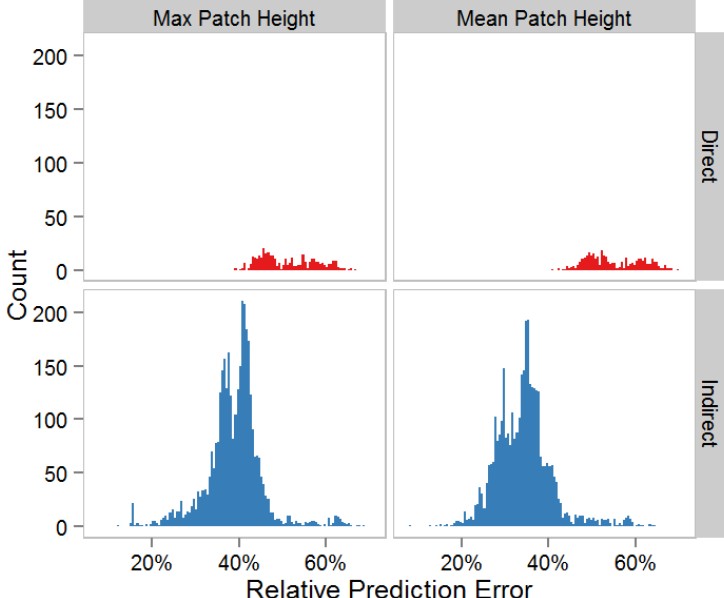

Figure 7. (a) Patch height and 95% prediction intervals (grey lines) for patches from direct
prediction and indirect prediction shown across the continuum of patch sizes. (b) Distributions of
relative prediction error (95% prediction interval) for patch height predictions.

**9    Tables**
Table 1. Summary of spaceborne datasets used to delineate or attribute forest patches.

| Dataset | Date | Attribute Value | Spatial Resolution |
|---|---|---|---|
| Landsat-7: ETM cloud-free composite; Vegetation Continuous Fields | c. 2013 | Top-of-atmosphere reflectance (mean): SWIR, NIR, Red, Green; Percent Tree Cover (mean) | 30 m pixel |
| HRSI: Worldview 1 & 2 | c. 2012 | DSM (mean, min, max, st. dev); NDVI (mean), Panchromatic roughness (mean); CRM (mean, st. dev) | ~ 0.5 m – 2 m pixel |
| ALOS PALSAR composite | 2007-2010 | backscatter power (HH, HV) | 25 m pixel |
| ICESat-GLAS LiDAR | 2003-2006 | ground surface elevation, waveform length | ~60 m diameter footprint |
