# Peer review of "Spaceborne potential for examining taiga-tundra ecotone form and vulnerability"

_Biogeosciences, 2015_

## Referee Comment (RC1) · Anonymous Referee #1 · 26 Feb 2016

This is a scientifically significant and useful investigation. The authors emphasise the idea that ecotone form (spatial characteristics) is an important way of understanding the tundra-taiga ecotone, and aim in this manuscript to develop the case that space-borne remotely sensed data have useful potential to characterise it, specifically by focussing on height distribution at the forest patch scale.

The MS is structured in an intelligible and expected way so it is straightforward to understand at least in principle how the authors have approached the question, though some details could benefit from more clarification. These are almost entirely within the methods section.

The abstract is clear and properly explains what the MS will do.

The introduction sections 1.1-1.2 provide good context to the study. Sections 1.3 and

1.4 may benefit from a little restructuring though: 1.3 starts by discussing general principles but then jumps to the implied conclusion that spaceborne LiDAR data will provide the necessary characterisation of height structure. Are there other (perhaps less promising) possibilities that should be discussed here, for example radar (or is this implied within the meaning of HRSI)? Section 1.4 is again general, so I think it logically belongs earlier than the decision to focus on the use of LiDAR data.

2.1 (study area) is fine. 2.2 (data acquisition and processing) is a bit hard to follow at times and needs more detail. Was the NDVI calculated from reflectance data or just from the uncalibrated pixel values of the HRSI data? And if the latter, were they atmospherically corrected first? How was the NDVI threshold determined? I think the processing to roughness needs some more information too. The approach used here is modelled on that used by Johansen et al (2014), but they were working with air orthophotos with a pixel size of 10 cm while the present work uses worldview imagery with pixels roughly ten times larger. How if at all do the different spatial and radiometric properties of the imagery affect the processing – e.g. choice of thresholds, kernel sizes? If different choices were made here than by Johansen et al, how were they informed? The rest of the methods section is clear.

Results are clear and interesting, and their discussion is comprehensive and sensible. The conclusions are properly justified.

Small details (by page/line number)

2/3 'asynchronous' – the word was unexpected here: you haven't said anything previously about structural changes being asynchronous, and I did not really understand what point you were making in using it.

4/1 'in the boreal' – the noun is missing!

4/8 'provide' → 'provides'

5/7 'Spaceborne uncertainty' isn't quite the right phrase, I think, since the uncertainty

hasn't originated in space. Maybe it needs a longer but more precise heading, such as 'uncertainty in spaceborne characterisation of TTE structure'?

5/10 'However. . . single active sensors. . .' I was a little puzzled by this phrasing. I don't think the work cited in the previous sentence uses exclusively active sensors (like LiDAR and radar), so am not sure what the 'however' is contradicting.

7/3 'sparse gradient in tree cover' = low gradient in tree cover, or sparse tree cover (or some combination of the two)?

7/26 'of primarily' → primarily of'

8/6 'backscatter power' → 'backscatter coefficient'

8/11 'DSM' I think this abbreviation is used here for the first time, so should be spelt out.

8/13 'attribute forest patches with the mean and variance. . .' This doesn't seem quite the right usage. Maybe you could say 'attribute the mean and variance. . . to the forest patches'.

8/29 'kernal' → 'kernel'

9/4 're-binned' → 'resampled'

9/19 'were filtered' → 'was filtered'

9/27 'attributing...with' – see 8/13

10/14 'attributed with' – again

15/24 superfluous 'the'. 'Theses' → 'These'

18/27 'describe' → 'describes'

22/3 'derived from a suite of. . .' → 'derived from a specific suite of. . .'

29 'backscatter power' → 'backscatter coefficient'

29 'scale' (in column heading) – would 'spatial resolution' be preferable?

33 figures (a) and (b) have been transposed.

---

## Referee Comment (RC2) · Anonymous Referee #2 · 7 Mar 2016

This paper tries to use multiple sources of remote sensing data particularly high resolution space borne imagery along with ALOS and GLAS LiDAR dataset to predict tree heights in taiga-tundra ecotone. This, if rigorously developed and clearly presented, may make significant contribution in understanding and modeling the horizontal and vertical heterogeneity in canopy heights, reducing the gap between remote sensing community and ecology community through datasets at the scale ecologists can use. This manuscript still needs clarification here and there to reach its potential for further interdisciplinary research.

The title of this manuscript is to examine the ecotone form and vulnerability. But the author did not specify or provide definitions in the paper what the form and vulnerability are (vulnerability was mentioned until the end of the manuscript). The form and vulnerability needs to be clearly specified in this study. For example, Page 3 line 20,

"recent work notes that rapid growth changes forms. . ." It is vague what the form here means. Does it refer to individual stand or patch scale increase in height? In some other places, it reads as the form of patch size and distribution. Additionally, the authors need to specify what factors the TTE may be vulnerable to.

Page 3 line 26-27, depending how extensive Taiga vegetation distributed, the height and relation with permafrost temperature actually varies (Roy-Levillee et al 2014). Double-check with the reference please.

Page 8 line 11, first time DSM is mentioned here, please spell out.

It seems that NDVI was used as a mask to determine whether the land cover is vegetated or not. It is not clear how the threshold was selected though. It will also be good to discuss/introduce roughness based on panchromatic HRSI image. Also discuss why this method can be useful without modification based on Johansen et al 2014.

For study region, the authors mentioned that the study area was exclusively covered by one single boreal species Larix gmelini. Please clarify if this is also the case for the verification and validation sites. It will be good to note what the tall shrub species/tundra plant communities are. This study looks at forest-tundra ecotone, but shrub species are just left out, which might also be tall and these may be the ones respond to warming and changes patch dynamics.

The Patch-based analysis sounds very straight forward and will reveal the local scale dynamics in TTE patches. However, it will be good to include a clear definition of patch as well. Maybe based on remote sensing texture characteristics "patch" seems to make sense. But how does it correlate to ecological meaning?
* * *

---

## Referee Comment (RC3) · Anonymous Referee #3 · 7 Mar 2016

This paper describes an analysis of the structure of the taiga-tundra ecotone (TTE) in north-central Siberia, using a combination of high resolution spaceborne imagery (HRSI), moderate resolution remote sensing, and spaceborne LiDAR. Their methodology includes a delineation of forest patch boundaries, in addition to both a direct estimate of forest patch heights, as well as an indirect modeling of forest patch heights. This approach has the capability of reducing the uncertainties involved with mapping the spatial structure of the TTE, for potential improvement of the land surface structure within earth system models. Generally, I found this paper to be quite good, and a nice contribution to the biogeosciences literature, specifically with regard to high latitude vegetation dynamics. Specific comments regarding scientific, methodological, and clarity issues: 1) In the second line of the Abstract (line 13), the authors use the term "asynchronous" to describe the fact that changes in vegetation structure can be

site-dependent, as well as circumpolar. I don't think that "asynchronous" is the best term to describe this phenomenon. 2) As the paper transitions from Introduction to Methods, the authors should state the objectives of the study much more clearly than they do. In the final paragraph of the Introduction, there is a "long-term goal," but that seems to be a goal for the scientific community, not necessarily for this study. Then there is a "short-term goal," which is to examine the uncertainty of mapped forest patch heights and to discuss the implications of this uncertainty. However, I think what the study actually does is more explicit than this short-term goal, i.e. maps forest patch distribution and develops remote sensing approaches to more accurately determine the heights of these patches – it does also address the uncertainty of these estimates. 3) "Non-forest" areas with mean roughness > 3 and mean NDVI < 0.25 were classified as forests. The authors may want to clarify what these forests actually look like. NDVI values of < 0.25 are very likely not indicative of forest vegetation. However, I can imagine that at the TTE, if the forest density was somewhat low within moderate resolution pixels, then it could be a patchy, low density forest with NDVI < 0.25. But, it might be a good idea to clarify this. I'm assuming this is not a mistake in the text. 4) It wasn't completely clear to me, but only patches > 0.5 ha had height estimates, yes? And, ∼90% of these were made using the indirect method, yes? 5) Probably my biggest concern with this paper is the inferences that are made with regard to monitoring of forest patch heights. One instance is the first line in the Discussion, but it occurs throughout the Discussion. The authors state that monitoring of forest structure (in this case patch height) "will help quantify the potential for changes in forest structure and… broader TTE dynamics," and "provide insight into the vulnerability to climate warming of current TTE structure." In my opinion, the leap from knowing the distribution of forest patch heights to assessing vulnerability to climate warming is a big one – it would be nice if the authors provided some further discussion of this inference. 6) On line 458, the authors state that "tree density is addressed with the delineation of forest patches." Tree density is addressed only coarsely, if at all. I don't think that there is any within-patch information on tree density here, unless I am mistaken – maybe from the LiDAR data?

Similarly (line 461), how is stem density quantified? 7) Lines 489-490 – Why does the current reported patch-level forest height uncertainty preclude an understanding of the most vulnerable portions of the TTE? Do we have any idea what are the most vulnerable portions of the TTE? 8) Lines 493-494 – Related to #5 above, how do these "general patterns of forest structure" suggest vulnerability and potential for changes? Again, the connection between the information provided in this study and the bigger picture of vegetation change and vulnerability is not well substantiated. Same for lines 525-526. Technical corrections: 1) Line 15 – "is" should be "are" 2) Line 45 – space between "2012)" and "and" 3) Line 63 – add "forest" after "boreal" 4) Line 85 – remove "s" from "resolves" 5) Line 100 – space between "scales" and "(Montesano" 6) Line 111 – remove "issues" 7) Line 145 – space between "isotherm" and "(Osawa" 8) Line 252 – move "both" after "specifying" 9) Line 317 – change "is" to "are" 10) Line 361 – remove "the" before "its modeling" 11) Line 393 – remove "s" from "features" 12) Line 394 – remove space between "present" and the period 13) Line 404 – remove "less than" 14) Line 507 – remove extra spaces between "estimates" and "provide" 15) Line 525 – remove "s" from "suggests"

---

## Referee Comment (RC4) · Anonymous Referee #4 · 10 Mar 2016

This paper presents how spaceborne remote sensing data (high resolution and medium resolution) can be used to predict Taiga-Tundra Ecotone (TTE) form and structure at a forest patch scale. The authors present a two-step methodology, by first estimating patch height directly from Lidar data, and then using these direct estimates in a random forest algorithm to predict patch height indirectly in the remaining patches. The uncertainty linked to these methods and their results are reported and examined in details. It is a very interesting work, highlighting the importance of such studies for environmental science. The combination of the individual trees / forest patches / coarser remote sensing data is definitely a very interesting approach, that could potentially be applied in more studies on forests around the globe. The paper is clearly written, easy to read and covers topics that are suitable for BGD. However, I would recommend some minor changes that would help improve the paper :

[Figure]

1) The paper is lacking some basic definitions and descriptions of terms the authors are using. How do they define terms such as "patches", "form", "vulnerability", "plot" vs. "stand". Issues related to TTE form determination are examined throughout the paper, but vulnerability is not directly addressed. The authors should make it clear from the beginning of the paper that vulnerability of forest patches can be directly linked to forest structure. This idea is suggested throughout the paper but is not stated clearly at the beginning. 2) One of the main conclusions of the study is that because the uncertainty is around 40%, remote sensing data, as presented in this paper, is not able to distinguish forest patches in terms of height or structure. Although this point is clear in the discussion and conclusion, it is not really covered in the abstract. 3) P.2, Line 3 : why "asynchronous"? Explain or remove from abstract. 4) The introduction is clear and interesting, but it would be nice to put the role of TTE into a more global perspective (how much do they represent, in terms of forest cover and/or biomass, why is it important to study them...) and to mention climate change and its impacts on TTE. 5) P.4, L.18-21 : Sentence is not clear. 6) The authors are using thresholds to mask or classify their remote sensing data, but do not explain how or why they picked these thresholds. What NDVI threshold did they use? Was that choice based on other studies? Why did they use a roughness threshold of 5.5? Same question for p.9, l.11. 7) p.7, l.11-14 : Ground reference data should be described in more details here. What kind of measurements have been made? Why are they outside of the selected sites? 8) p.8, l. 11 : define DSM (definition given p.9). 9) P.9, l.15 : mention GLAS footprint size and explain why you used a radius of 10m (l.23). 10) P.12, l8-10 : I find this sentence and Fig 3b misleading. The fact that the sampling density is higher in smaller patches is simply due to the fact that the authors only selected the patches that had GLAS shots in them, hence giving a higher number of samplings per ha in small patches. The reader should be reminded of this fact here. Adding the average and maximum number of samples per patch in each class would give a better idea of the distribution of samples, in addition to figure 3b. 11) P.12, l.15 and figure 4 : what do you mean by plot/stand? 12) P.12, paragraph 3.2 : a) Why are the ground data plots

outside of the selected sites? Does it make a difference? b) Why are the calibration and validation sites separated spatially? Are the two areas similar in terms of topography, forest structure...? Wouldn't it be better and less biased to select them randomly for calibration or validation? 13) p.17, Discussion : The authors could mention future spaceborne missions, such as GEDI, and the possibilities they would bring for this kind of studies. 14) P.17, l.14-17 : Sentence is not clear. Reformulate. 15) Did the authors take the shape of GLAS footprints into account? GLAS footprint is not always exactly a circle of 60m diameter and these differences might have an impact on the results, if not taken into account. 16) P.19, l.19-13. Not clear, reformulate.

Comments about figures : Figure 1 : Why are the study sites so far away from the ground reference sites? Their height and structure characteristics might be different than the ones of the study sites. Figure 3 : a) I recommend to normalize the histograms, to make the two datasets more comparable. Instead of # of forest patches, show frequency (# / total # of each dataset). b) see comment 10). Figure 4 : a) and b) do not match caption. a) : see comment 12b. b) Normalize histograms. c) In caption, add "50th, and 75th percentile of mean height" for clarity. Figure 7 b) Normalize histograms It would be much easier to compare the direct and indirect histograms if they were all normalized.

Specific comments : 1) p.2, l.2 : "changes" instead of "change", or "occurs" instead of "occur". 2) P.4, l.24 : comma is not necessary : "group of trees, may help". 3) P.5, l.2 : remove "and" in "biodiversity, and biogeochemical". 4) P.5, l.26 : replace "," by "." In "structure, however". 5) P.11, l.11 : remove "the" in "specifying the both number". 6) P.19, l.9 : "explains" instead of "explain". 7) P.22, l. 9 : "suggest" instead of "suggests"
* * *

---

## Author Comment (AC1) · 11 Apr 2016

P. M. Montesano

paul.m.montesano@nasa.gov

Responses to RC1

Comment 1. Sections 1.3 and 1.4 may benefit from a little restructuring though: 1.3 starts by discussing general principles but then jumps to the implied conclusion that spaceborne LiDAR data will provide the necessary characterisation of height structure. Are there other (perhaps less promising) possibilities that should be discussed here, for example radar (or is this implied within the meaning of HRSI)? Section 1.4 is again general, so I think it logically belongs earlier than the decision to focus on the use of LiDAR data.

Response: We agree that these two sections can benefit from some minor restructuring. We will clarify that the approach to which we refer involves is a general multi-sensor

one that includes passive and active remote sensing from multispectral imagery, LiDAR and SAR at a variety of spatial resolutions. We did not intend to get specific in section 1.3, but rather point out that a patch-level approach that incorporates data from a range of sensor types may help capture both vertical and horizontal TTE patterns. To this end, our edits will provide a cleaner transition between Sections 1.3 and 1.4.

Comment 2. 2.2 (data acquisition and processing) is a bit hard to follow at times and needs more detail. Was the NDVI calculated from reflectance data or just from the uncalibrated pixel values of the HRSI data? And if the latter, were they atmospherically corrected first? How was the NDVI threshold determined? I think the processing to roughness needs some more information too. The approach used here is modelled on that used by Johansen et al (2014), but they were working with air orthophotos with a pixel size of 10 cm while the present work uses worldview imagery with pixels roughly ten times larger. How if at all do the different spatial and radiometric properties of the imagery affect the processing – e.g. choice of thresholds, kernel sizes? If different choices were made here than by Johansen et al, how were they informed? The rest of the methods section is clear.

Response: We agree that this section can benefit from a bit more detail. Our approach for separating vegetation and non-vegetation was to use NDVI calculated on uncalibrated digital number values of pixels and a threshold determined from a sample of vegetation and non-vegetation patches to provide a preliminary veg/non-veg mask. This preliminary mask was modified with image roughness information to identify forest from non-forest vegetation. Our approach involving image roughness is resolution independent in that feature roughness can be captured as long as those features are resolved in the imagery. Johansen et al. use 10cm data to identify individual banana plant leaves, while we use ∼60cm data to capture groups of larch trees. This methodology captures image texture derived from variations in image brightness that is a result of the arrangement of trees across the landscape. An exhaustive examination of the influence of varying (1) kernel sizes, (2) image radiometry, and (3) thresholds on the

identification of forest patches was not explored in this study. We will add mention of this in the Methods section. See also Comment #6 from responses to Reviewer #4.

Small details (by page/line number) 2/3 'asynchronous' – the word was unexpected here: you haven't said anything previously about structural changes being asynchronous, and I did not really understand what point you were making in using it.

Response: We will remove this term from the abstract.

4/1 'in the boreal' – the noun is missing! 4/8 'provide' → 'provides'

Response: These changes will be made.

5/7 'Spaceborne uncertainty' isn't quite the right phrase, I think, since the uncertainty hasn't originated in space. Maybe it needs a longer but more precise heading, such as 'uncertainty in spaceborne characterisation of TTE structure'?

Response: We will consider rewording this heading in the next version of the manuscript.

5/10 'However. . . single active sensors. . .' I was a little puzzled by this phrasing. I don't think the work cited in the previous sentence uses exclusively active sensors (like LiDAR and radar), so am not sure what the 'however' is contradicting.

Response: We agree that this sentence is poorly worded and will be re-worked in the next version.

7/3 'sparse gradient in tree cover' = low gradient in tree cover, or sparse tree cover (or some combination of the two)?

Response: We agree that this term is not clear. We should say, as do our references, that this region features open or sparse tree cover.

7/26 'of primarily' → primarily of' 8/6 'backscatter power' → 'backscatter coefficient' Response: These data were in power units (0-1). 8/11 'DSM' I think this abbreviation

is used here for the first time, so should be spelt out. 8/13 'attribute forest patches with the mean and variance. . .' This doesn't seem quite the right usage. Maybe you could say 'attribute the mean and variance. . . to the forest patches'. 8/29 'kernal' → 'kernel' 9/4 're-binned' → 'resampled' 9/19 'were filtered' → 'was filtered' 9/27 'attributing...with' – see 8/13 10/14 'attributed with' – again 15/24 superfluous 'the'. 'Theses' → 'These' 18/27 'describe' → 'describes' 22/3 'derived from a suite of. . .' → 'derived from a specific suite of. . .' 29 'backscatter power' → 'backscatter coefficient' Response: See above 29 'scale' (in column heading) – would 'spatial resolution' be preferable? 33 figures (a) and (b) have been transposed.

Response: The edits suggested above will be made unless otherwise noted.

––––––––––––––––––––––––––––––

---

## Author Comment (AC2) · 11 Apr 2016

Responses to RC2

Comment 1. The title of this manuscript is to examine the ecotone form and vulnerability. But the author did not specify or provide definitions in the paper what the form and vulnerability are (vulnerability was mentioned until the end of the manuscript). The form and vulnerability needs to be clearly specified in this study. For example, Page 3 line 20, "recent work notes that rapid growth changes forms. . ." It is vague what the form here means. Does it refer to individual stand or patch scale increase in height? In some other places, it reads as the form of patch size and distribution. Additionally, the authors need to specify what factors the TTE may be vulnerable to.

Response: The Reviewer suggests a the need to more clearly define form and vulner-

ability up front, and points out that a vague reference to form appears (pg 3, line 20) before it is defined (pg 4, line 16). We will fix these incoherences in the next version.

Comment 2. Page 3 line 26-27, depending how extensive Taiga vegetation distributed, the height and relation with permafrost temperature actually varies (Roy-Levillee et al 2014). Double-check with the reference please

Response: The Reviewer is correct and we will edit the manner in which we reference that study to more accurately reflect that the variation in permafrost temperature is controlled in part by vegetation height, but also by the arrangement of taiga patches.

Comment 3. Page 8 line 11, first time DSM is mentioned here, please spell out.

Response: Page 8 line 11: We will insert 'digital surface model' here before 'DSM'

Comment 4. It seems that NDVI was used as a mask to determine whether the land cover is vegetated or not. It is not clear how the threshold was selected though. It will also be good to discuss/introduce roughness based on panchromatic HRSI image. Also discuss why this method can be useful without modification based on Johansen et al 2014.

Response: RC2 requests more information on the NDVI threshold used to separate vegetation from non-vegetation. Please see Comment #2/Response to Reviewer #1 and Comment #6/Response to Reviewer #4.

Comment 5. For study region, the authors mentioned that the study area was exclusively covered by one single boreal species Larix gmelini. Please clarify if this is also the case for the verification and validation sites. It will be good to note what the tall shrub species/tundra plant communities are. This study looks at forest-tundra ecotone, but shrub species are just left out, which might also be tall and these may be the ones respond to warming and changes patch dynamics.

Response: Both the study region where forest patches were mapped and the verification and validation sites featured the same forest type; exclusively Larix gmelini. We

think the Reviewer makes a good point in suggesting we include some information on tall shrub species and tundra communities. We will add this information to the Study Area section. We will also note that we do not directly address shrub structure, as our field data do not include shrub measurements. However, our remote sensing analysis may include tall shrubs that may persist within the forest mask, and thus, a component of the patch height and uncertainty estimates may include shrub information. This warrants mention in the Discussion.

Comment 6. The Patch-based analysis sounds very straight forward and will reveal the local scale dynamics in TTE patches. However, it will be good to include a clear definition of patch as well. Maybe based on remote sensing texture characteristics "patch" seems to make sense. But how does it correlate to ecological meaning?

Response: Here we use the term 'forest patch' to refer to a group of trees that are relatively homogenous in terms of height and consistent in terms of horizontal arrangement. We will include text in Section 1.2 to clearly state what we mean by 'forest patch': The spatial configuration of tree of similar structure can be conceptualized as 'forest patches', whereby a patch represents a group of trees that exhibit relatively homogenous or consistent vertical and horizontal structure.

---

## Author Comment (AC3) · 11 Apr 2016

Responses to RC4

Comment 1. The paper is lacking some basic definitions and descriptions of terms the authors are using. How do they define terms such as "patches", "form", "vulnerability", "plot" vs. "stand". Issues related to TTE form determination are examined throughout the paper, but vulnerability is not directly addressed. The authors should make it clear from the beginning of the paper that vulnerability of forest patches can be directly linked to forest structure. This idea is suggested throughout the paper but is not stated clearly at the beginning.

Response: We agree the next version should more clearly define some of these terms. We point out that 'form' is defined in Section 1.2. In Section 3.2 we will clarify the a

plot is a 15m in radius while stands are derived from Bondarev et al. 1997. We will introduce 'patch' in Section 1.2 (see Reviewer #2 Comment #6/Response). We'd like to introduce 'vulnerability' in at the end of Section 1.2 in the following manner: "Epstein et al. 2004 provide a synthesis of how TTE dynamics and patterns are linked, and that a better understanding of vegetation transitions can improve predictions of vegetation sensitivity. Their observations provide a basis for the inference that TTE structure is most vulnerable to temperature-induced changes in structure where its structure is temperature-limited. Vulnerable portions of the TTE are areas most likely to experience changes in forest structure that alter TTE structural patterns."

Comment 2. One of the main conclusions of the study is that because the uncertainty is around 40%, remote sensing data, as presented in this paper, is not able to distinguish forest patches in terms of height or structure. Although this point is clear in the discussion and conclusion, it is not really covered in the abstract.

Response: We will update the abstract to better align with the point as it is made in other sections.

Comment 3. P.2, Line 3 : why "asynchronous"? Explain or remove from abstract.

Response: We will remove this from the abstract to avoid confusion.

Comment 4. The introduction is clear and interesting, but it would be nice to put the role of TTE into a more global perspective (how much do they represent, in terms of forest cover and/or biomass, why is it important to study them. . .) and to mention climate change and its impacts on TTE.

Response: We will add a point to the Introduction mentioning the global importance of the TTE.

Comment 5. P.4, L.18-21 : Sentence is not clear.

Response: We will reword this sentence to clarify.

Comment 6. The authors are using thresholds to mask or classify their remote sensing data, but do not explain how or why they picked these thresholds. What NDVI threshold did they use? Was that choice based on other studies? Why did they use a roughness threshold of 5.5? Same question for p.9, l.11.

Response: The thresholds for both NDVI and roughness used to classify forest were based on preliminary interpretation and sampling of these image layers for forest and non-forest areas across all forest patch mapping sites. The goal of this preliminary explorative work was to understand the range of roughness and NDVI values that indicated forest. This explorative work identified thresholds that were image independent and could be used in an automated patch classification protocol. While used in an image independent manner across study sites, these thresholds are sensitive to the seasonality of vegetation and, likely, the sun-sensor-target geometry at which the imagery was acquired. A more in-depth examination of how the distribution of NDVI and roughness varied for forest patches across different images was not part of this work. We can note this in the next version of the manuscript. See also Comment #2/Response to Reviewer #1.

Comment 7. p.7, l.11-14 : Ground reference data should be described in more details here. What kind of measurements have been made? Why are they outside of the selected sites?

Response: We provided reference to a paper (Montesano et al .2014) where ground data collection was described in more detail in a previous. However, we agree that it may be helpful to provide a bit more detail. In summer 2008, we measured tree diameters at breast height (DBH, 1.3 m) and tree heights (clinometers for 97% of trees and tape measurement for 3%) at plots coincident with GLAS LiDAR footprints. The data used for this study included DBH for all tree stems with DBH >3 cm ($\pm$0.1 cm) and corresponding tree heights for each tree in each plot. These plot data represented a range of sparse Larix forest conditions found across northern Siberia Larix forests, excluding prostrate Larix forms. The forest mapping sites do not spatially coincide with

our ground plots because this study aims to examine the TTE on the Kheta-Khatanga Plain which exhibits a range of TTE forms, where the TTE covers a broader area, and where we had access to both stereo and multispectral HRSI data. While not spatially coincident, our ground plots characterize very similar forest conditions - the main difference being the ecotone is compressed (covers a smaller area) in the region of our ground data due to topography. The forest type (Larix gmelini) and structure is consistent across the broader region (see stand data from regionally distributed sites in Bondarev 1997).

Comment 8. p.8, l. 11 : define DSM (definition given p.9). Comment 9. P.9, l.15 : mention GLAS footprint size and explain why you used a radius of 10m (l.23).

Responses to 8,9: These changes will be made. GLAS footprints were approximated with ∼60m diameter footprints. The 10m radius was used as part of a filtering procedure to include GLAS footprints that were coincident with DSM elevation measurements that would be able to capture forest heights where trees are often < 12m. This radius helped remove footprints for which there was a broad range of DSM values near the footprint centroid that was indicative of terrain slope interfering with height estimates.

Comment 10. P.12, l8-10 : I find this sentence and Fig 3b misleading. The fact that the sampling density is higher in smaller patches is simply due to the fact that the authors only selected the patches that had GLAS shots in them, hence giving a higher number of samplings per ha in small patches. The reader should be reminded of this fact here. Adding the average and maximum number of samples per patch in each class would give a better idea of the distribution of samples, in addition to figure 3b.

Response: We report the density of LIDAR samples for the set of patches whose height was sampled with LiDAR (directly). So, within this group (defined explicitly as being sampled with LiDAR), the smaller patches will have higher sampling density (but not necessarily more samples). The violin plots demonstrate the distribution of sampling

densities for each general forest patch size group for which direct height measurements (using LiDAR) were made.

Comment 11. P.12, l.15 and figure 4 : what do you mean by plot/stand?

Response: This will be clarified. Trees measurements described in Montesano et al. 2014 are associated with the term 'plots' while 'stands' is the term used by Bondarev 1997.

Comment 12. P.12, paragraph 3.2 : a) Why are the ground data plots outside of the selected sites? Does it make a difference? b) Why are the calibration and validation sites separated spatially? Are the two areas similar in terms of topography, forest structure. . .? Wouldn't it be better and less biased to select them randomly for calibration or validation?

Response: (a) See response to Comment #7 (b) Figure 4b,c summarize the forest structure across all calibration and validation sites showing the range of tree heights measured in the field.

Comment 13. p.17, Discussion : The authors could mention future spaceborne missions, such as GEDI, and the possibilities they would bring for this kind of studies.

Response: We will note in section 4.3 that future spaceborne missions will provide more ground surface elevation samples needed for improving patch height estimates. ICESat-2 will be useful for the TTE, as GEDI will only sample below ∼50N.

Comment 14. P.17, l.14-17 : Sentence is not clear. Reformulate. Comment 15. Did the authors take the shape of GLAS footprints into account? GLAS footprint is not always exactly a circle of 60m diameter and these differences might have an impact on the results, if not taken into account.

Responses to 14,15: After Montesano et al. 2014, we used a 10m radius circle centered on GLAS footprint centroids to capture DSM surface elevations. Because we focus on DSM elevation data near the centroid, the precise shape of the footprint (which

is actually an ellipse) will not influence results.

16. P.19, l.19-13. Not clear, reformulate.

Response: We will clarify the link between horizontal structure and image texture.

Comments about figures : Figure 1 : Why are the study sites so far away from the ground reference sites? Their height and structure characteristics might be different than the ones of the study sites.

Response: See response to Comment #7

Figure 3 : a) I recommend to normalize the histograms, to make the two datasets more comparable. Instead of # of forest patches, show frequency (# / total # of each dataset). b) see comment 10).

Response: (a) We argue that it is more helpful to show the y-axis with absolute counts of forest patches (b) See response to Comment #10

Figure 4 : a) and b) do not match caption. a) : see comment 12b. b) Normalize histograms. c) In caption, add "50th, and 75th percentile of mean height" for clarity. Figure 7 b) Normalize histograms It would be much easier to compare the direct and indirect histograms if they were all normalized.

Response: We will switch the captions to match the figures and add "..percentile of mean height..' as suggested. We appreciate the suggestions to normalize histograms but we argue that showing actual numbers of forest patches per bin is easier to understand because it highlights the overall quantity of patches receiving indirect height estimates as compared to those receiving direct estimates.

Specific comments : 1) p.2, l.2 : "changes" instead of "change", or "occurs" instead of "occur". 2) P.4, l.24 : comma is not necessary : "group of trees, may help". 3) P.5, l.2 : remove "and" in "biodiversity, and biogeochemical". 4) P.5, l.26 : replace "," by "." In "structure, however". 5) P.11, l.11 : remove "the" in "specifying the both number". 6)

P.19, l.9 : "explains" instead of "explain". 7) P.22, l. 9 : "suggest" instead of "suggests"

Response: These changes will be included in the next version of the manuscript.

---

## Author Comment (AC4) · 11 Apr 2016

Responses to RC3

Comment 1. In the second line of the Abstract (line 13), the authors use the term "asynchronous" to describe the fact that changes in vegetation structure can be site-dependent, as well as circumpolar. I don't think that "asynchronous" is the best term to describe this phenomenon.

Response: We will remove this term from the abstract.

Comment 2. As the paper transitions from Introduction to Methods, the authors should state the objectives of the study much more clearly than they do. In the final paragraph of the Introduction, there is a "long-term goal," but that seems to be a goal for the

scientific community, not necessarily for this study. Then there is a "short-term goal," which is to examine the uncertainty of mapped forest patch heights and to discuss the implications of this uncertainty. However, I think what the study actually does is more explicit than this short-term goal, i.e. maps forest patch distribution and develops remote sensing approaches to more accurately determine the heights of these patches – it does also address the uncertainty of these estimates.

Response: We agree that the objectives can be stated more clearly. We will clarify the specific objectives of this paper, and explain how they fit with longer term scientific objectives for examining forest structure change in the TTE.

Comment 3. "Non-forest" areas with mean roughness > 3 and mean NDVI < 0.25 were classified as forests. The authors may want to clarify what these forests actually look like. NDVI values of < 0.25 are very likely not indicative of forest vegetation. However, I can imagine that at the TTE, if the forest density was somewhat low within moderate resolution pixels, then it could be a patchy, low density forest with NDVI < 0.25. But, it might be a good idea to clarify this. I'm assuming this is not a mistake in the text.

Response: We agree that clarification is needed, and that the description as it exists now is confusing. Due to the iterative nature of the classification, initial classification steps provide temporary classes that are refined with subsequent classification steps. Please see Comment #2/Response to Reviewer #1 and Comment #6/Response to Reviewer #4.

Comment 4. It wasn't completely clear to me, but only patches > 0.5 ha had height estimates, yes? And, âĹij90% of these were made using the indirect method, yes?

Response: Correct, the minimum mapping unit (patch size) was set at > 0.5 ha. and 90% of these patches featured height estimates that were derived indirectly. We can adjust our wording of this.

Comment 5. Probably my biggest concern with this paper is the inferences that are

made with regard to monitoring of forest patch heights. One instance is the first line in the Discussion, but it occurs throughout the Discussion. The authors state that monitoring of forest structure (in this case patch height) "will help quantify the potential for changes in forest structure and. . . broader TTE dynamics," and "provide insight into the vulnerability to climate warming of current TTE structure." In my opinion, the leap from knowing the distribution of forest patch heights to assessing vulnerability to climate warming is a big one – it would be nice if the authors provided some further discussion of this inference.

Response: The Reviewer expresses concern with portions of the Discussion where inferences are made with respect to the monitoring of forest patch height. This concern may arise from some wording we use to describe the link between forest structure patterns and vulnerability to structural changes. We note the Reviewer's concern, and plan to modify the first paragraph of the Discussion to reflect the following: Recent literature suggests that TTE form, or pattern, may reflect which portions of the TTE are controlled primarily by temperature. With remote sensing, TTE forms/patterns can be identified by characterizing the horizontal and vertical structure of trees. By identifying these forms, TTE controls may be inferred. The ability to characterize horizontal and vertical structure is a precursor to both (1) distinguishing one TTE form/pattern from another, and (2) identifying areas where TTE form/pattern suggests tree growth is temperature limited. The intersection of such temperature limited TTE form/pattern with regional warming trends may point to areas where TTE structure is vulnerable to changes in structure. Our work demonstrates the potential from spaceborne remote sensing for depicting a key structural characteristic of TTE form (height), and suggests where improvements are needed in order to identify portions of the TTE vulnerable to warming-induced structural changes. Alsoc, see Comment #1/Response to Reviewer #4.

Comment 6. On line 458, the authors state that "tree density is addressed with the delineation of forest patches." Tree density is addressed only coarsely, if at all. I don't think

that there is any within-patch information on tree density here, unless I am mistaken – maybe from the LiDAR data? Similarly (line 461), how is stem density quantified?

Response: The Reviewer points out that tree/stem density is addressed in a coarse manner, and asks how stem density is quantified. We agree with the Reviewer that stem density is coarsely addressed. However, we indicate that we use image roughness/texture as a general proxy for horizontal vegetation structure, which includes tree/stem density. Image texture measures have been used to examine horizontal forest struture (e.g., Wood et al. 2012; http://www.sciencedirect.com/science/article/pii/S0034425712000156, Wood et al. 2013; http://journals.plos.org/plosone/article?id=10.1371/journal.pone.0063211)

Comment 7. Lines 489-490 – Why does the current reported patch-level forest height uncertainty preclude an understanding of the most vulnerable portions of the TTE? Do we have any idea what are the most vulnerable portions of the TTE?

Response: The Reviewer identifies an insufficient explanation as a source of confusion regarding the link between patch height uncertainty and the identification of temperature-limited portions of the TTE. It would be helpful if we more clearly define our terms. Please see Comment #1/Response to Reviewer #4.